# Age patterns in subjective well-being are partially accounted for by psychological and social factors associated with aging

Arthur A. Stone[1,2]*, Joan E. Broderick[2], Diana Wang[2], Stefan Schneider[2]

**1** Department of Psychology, University of Southern California, Los Angeles, California, United States of America, **2** Dornsife Center for Self-Report Science, University of Southern California, Los Angeles, California, United States of America

* arthuras@usc.edu

## Abstract

Subjective well-being has captured the interest of scientists and policy-makers as a way of knowing how individuals and groups evaluate and experience their lives: that is, their sense of meaning, their satisfaction with life, and their everyday moods. One of the more striking findings in this literature is a strong association between age and subjective well-being: in Western countries it has a U-shaped association over the lifespan. Despite many efforts, the reason for the curve is largely unexplained, for example, by traditional demographic variables. In this study we examined twelve social and psychological variables that could account for the U-shaped curve. In an Internet sample of 3,294 adults ranging in age from 40 to 69 we observed the expected steep increase in a measure of subjective well-being, the Cantril Ladder. Regression analyses demonstrated that the social-psychological variables explained about two-thirds of the curve and accounting for them significantly flattened the U-shape. Perceived stress, distress-depression, an open perspective about the future, wisdom, satisfaction with social relationships, and family strain were measures that had pronounced impacts on reducing the curve. These findings advance our understanding of why subjective well-being is associated with age and point the way to future studies.

## Introduction

Subjective well-being (SWB) is a major concept of interest for augmenting other economic and social measures of welfare [1, 2]; it includes the concepts of perceived meaning in life, evaluations of satisfaction with life, and everyday moods. As such, it has received much attention over the last two decades (for example, by the OECD, WHO, and many governments). A consensus among behavioral scientists and economists is that measures of subjective well-being are associated with age and, yet, the patterns of association are not entirely consistent with everyday beliefs about aging [3–6]. Because health deteriorates over age, a reasonable speculation is that peoples' subjective well-being would also decline. However, dozens of cross-sectional and longitudinal studies have shown in the US and Western Europe that SWB *increases* in older age (around age 50), which may reasonably be considered a paradox. Moreover, this

**Data Availability Statement:** Data files and documentation are available from the Open Science Framework database (URL: osf.io/ufmgh).

**Funding:** AAS and JEB received funding from Princeton Center for Translational Research on Aging and from the National Institute on Aging Grant AG042407. The funders had no role in study design, data collection and analysis, decision to publish, or preparation of the manuscript.

**Competing interests:** AAS is a Senior Scientist with the Gallup Organization and is a consultant for Adelphi Values. This does not alter our adherence to PLOS ONE policies on sharing data and materials.

pattern of a U-shaped association with a dip in midlife is seen across many countries around the world [7]. In this paper we consider the association between age and one of the prime measures of subjective well-being, the Cantril Ladder. We focus on an age range from 40 to 69 years to capture the theoretically expected nadir and subsequent age-related increase in SWB. We examine a number of theoretically relevant variables, many derived from a life-span perspective, that could possible explain the higher levels of SWB in older age.

Apart from declining health, aging is replete with biological development and social changes associated with the lifespan, and many of these changes are natural candidates for explaining the improvement in SWB after age 50. Age-related changes in family composition, employment status, social networks, physical capabilities, and income—just to name a few—appear capable of exerting influence on SWB. Although many such variables have been examined, none have emerged as definitive explanations of the age-SWB curves. For example, using several years of the Gallup Organization's daily poll, an extensive telephone interview conducted with hundreds of thousand people in the US every year, we tested four standard demographic variables and were unable to explain the age-associated curves [8]. That is, after appropriate statistical controls for the possible explanatory variables, the curve remained. The explicit testing of potential explanatory variables is necessary, because it does not logically follow that variables exhibiting the same or similar age curves as subjective well-being *necessarily explain* those curves; that is, a variable that is solely associated with age does not necessarily account for the age-Cantril Ladder curve.

The reasons why the Ladder has this age pattern remain unknown. In addition to advancing life-span theories of subjective well-being, understanding the drivers of the pattern could have policy implications for improving SWB across the adult life span. Apart from the variables mentioned above, it has been conjectured that more subtle alterations in psychological and social functioning, including accumulated knowledge or shifts in worldviews, may explain age-related changes in SWB [3]. Perhaps most influential in this regard is Carstensen's theory of aging (Socioemotional Selectivity Theory, SST; [9]) that advances the proposition that social-psychological transitions over age improve SWB in older age. The current study tested twelve psychological and social variables that we posit could account for the age-Ladder relationship (hereafter referred to as *explanatory variables*). They were culled from a review of the literature on SWB and aging [2, 4, 10–12]. We required that all measures were able to be administered via an online survey to an Internet-based panel or modified them so that they could be. Furthermore, in order to increase the likelihood that they could account for age differences in the Ladder, we chose variables that had known associations with age. Descriptions of the measures are shown in Table 1.

In summary, the findings show that a number of the psychological and social explanatory variables partially examined here accounted for the U-shaped association between age and subjective well-being. To our knowledge, these are the first results to empirically account for the curve, with the caveats that the curve was only partially eliminated and that the cross-section nature of the study precludes causal conclusions. Nevertheless, the results provide support for the theories from which these variables were derived and suggest more extensive, prospective studies of their association with subjective well-being.

## Materials and methods

### Sample

Participants in our sample were recruited from a multi-source panel through Dynata Inc. The inclusion criteria for the study include residence in the United States and the use of a computer or desktop (rather than mobile device or tablet). Adults were recruited in the US

**Table 1. Description of Cantril Ladder and potential explanatory variables.**

| Variable | Description and Source |
|---|---|
| Cantril Ladder | Subjective well-being was measured using the Cantril Ladder. Participants viewed an image of a ladder with 11 steps (0—worst possible life, 10—best possible life) and reported which step best represents their life [13] |
| Perceived Stress (PSS) | Perceived stress is measured using a 4-item short-form of the Perceived Stress Scale, in which participants are about the extent to which they felt stressed over the last month ranging from 1 (never) to 5 (very often) [14] |
| Future time perspective *as open* (FTP-Open) | The perception of one's future time as open-ended and holding opportunities is measured with the 4-item 'Open' dimension of the Future Time Perspective Brief Multidimensional Scale ranging from 1 (strongly disagree) to 5 (strongly agree) [15] |
| Future time perspective *as limited* (FTP-Limited) | The perception of one's future as limited and closed in nature is measured with the 4-item 'Limited' dimension of the FTP Brief Multidimensional scale [16] |
| Social network size (Network Size) | Participants were asked for the number of ties in each of three concentric circles which represent the level of closeness and importance of the relationships. The Social Convoy Circles was adapted to a web-based format [17] |
| Social satisfaction | Participants rated how satisfied they were in general with their social partners and family/relatives on a rating scale ranging from 1 (very dissatisfied) to 5 (very satisfied) [18] |
| Social comparison | The tendency to compare oneself with others, or an orientation toward comparison is measured using an 11-item Iowa Netherlands Comparison Orientation Measure on a five-point scale ranging from 1(strongly disagree) to 5(strongly agree) [19] |
| Positivity Effect | The positivity effect refers to a greater memory for positive over negative information in older age. An experimental task was adapted to a web-based format [20, 21] wherein participants were presented with positive, negative, and neutral images from the International affective picture system (IAPS; [22]) in an initial phase of the task. After a short delay, recognition memory is tested with a presentation of previously-shown images as well as new ones. Accuracy of memory for *positive* images is calculated by subtracting ratio of incorrectly recognized images from the ratio of correctly recognized photos. |
| Relationship quality: Support *from family* | Relationship quality is assessed with support and strain dimensions from the support and strain questionnaire. Support is measured by four items that assess the extent to which individuals feel that others care for them and are able to help. All items are answered on a 5-point Likert-type scale ranging from 1 (disagree) to 5 (agree) [23] |
| Relationship quality: Support *from friends* | Same as above but for friends. |
| Relationship quality: Strain *from family* | Strain is measured by two items that assess the extent to which participants feel that their family members make too many demands or get on their nerves. All items are answered on a 5-point Likert-type scale ranging from 1 (disagree) to 5 (agree). |
| Relationship quality: Strain *from friends* | Same as above but for friends. |
| Wisdom | Wisdom is measured with a 12-item abbreviated three-dimensional wisdom scale (3D-WS-12; [24]): cognitive (an understanding of life), reflective (ability to look at phenomena from different perspectives), and affective (presence of positive emotions toward others). All items are on a five-point scale ranging from 1 (strongly disagree) to 5 (strongly agree). |
| Distress-Depression | Depression is measured using a 4-item short form measure of emotional distress-depression including items such as 'I felt depressed' or 'hopeless' over the past 7 days (PROMIS; [25]). All items are on a 5-level response scale ranging from 1 (never) to 5 (always). |

between the ages of 40 and 69, a period during which SWB increases. Prior work has shown the greatest increase in Cantril Ladder scores in this age range. Given limited resources for sample recruitment, we focused on this age range to obtain maximum statistical power with this sample size.

Sociodemographic criteria were set to fill predetermined quota of evenly distributed genders, age groups (40–49, 50–59, 60–69), and race/ethnicity that corresponded to Census breakdown (64% White/Caucasian not Hispanic, 12% Black/African American, 16% Hispanic, and 8% other). The survey was administered between May 28th and June 11th, 2019. Potential participants in the Dynata panel can qualify for studies based on known demographic or other characteristics. Screener questions are presented to panelists for a batch of studies in a single work session. Panelist qualify for multiple surveys in a single session, which reduces screen-outs and the temptation to cheat to qualify for surveys, and surveys are adjusted for length to reduce fatigue. Participants are rewarded upon completion of the survey with points which can be redeemed for a variety of predetermined forms (cash, donation to charity etc.).

## Internet survey

The questionnaire was designed and administered to participants using Qualtrics. The study was approved by the Institutional Review Board at the University of Southern California (UP-19-00314). The average time of completion of the survey was 33.6 minutes, ranging from 7.8 to 5844 minutes.

## Analyses

Age is considered as a continuous variable in all analyses. In order to provide maximal information about associations between age and the Cantril Ladder we visualize the associations using locally weighted scatterplot smoothing (Lowess), which will allow us to see any nonlinear associations (which we expect from prior work). Inferential testing of curvilinear, associations was accomplished with ordinary least squares regression incorporating Age and Age squared as independent variables. The exception was that household income was retained as a continuous variable.

To test the hypothesis that the candidate variables account for the some or all of age associations with the Cantril Ladder, we use a technique reported in our prior papers wherein two models are compared and regression coefficients common to both models are tested for equivalence. In this case, the first model predicts the Ladder from a reduced model (e.g., age variables) and the second model includes the terms from the first model *and* a set of new variables. The hypothesis of significantly changed age regression coefficients is then tested with STATA's Seeming Unrelated Estimation Testing (SUEST). Confirmation of the hypothesis would be that the regression coefficients are significantly different in the two models *and* that those changes result in a reduced association between age and the Ladder, which would graphically be shown by a relatively flatter association.

Four regression models are used in this process. The first examines linear and curvilinear components of age (M1), the second adds a set of three relatively stable demographic variables (M2: gender, education, and race), the third adds a set of two demographic variables that are likely to change over the lifespan (M3: income and marital status), and the fourth adds the set of 13 candidate variables described above (M4). Dichotomous coding of the demographic variables is consistent with our prior work, however, given the potential importance of income, we have coded this with 9 levels. To quantify the magnitude of the effects of the explanatory variables, we examined the extent (i.e., proportion) to which the linear regression coefficient of age on the Cantril Ladder was reduced by inclusion of an explanatory variable as covariate

[26]. In these models each explanatory variable was considered individually in separate regression analyses that controlled for all demographic variables. Because the quadratic age term was also controlled, we centered age at 56 years (the median age in the sample) and obtain the proportion of the linear age effect reduced by an explanatory variable at that age.

## Results

### Cantril Ladder and age

A final sample of 3,294 was obtained and about half of the sample was female (52%), most were White (including White Hispanic/Latino; 78%), over half completed a college degree (53%), almost two-thirds were married or cohabiting (63%), and the median household income fell into the category $50,000-$74,999 (Table 2). The average Cantril Ladder score was 6.82 (for comparison: it was 6.60 for those aged 40–69 in the 2008 Gallup survey [8]). Regression analyses showed that Ladder scores were associated with age (linear and quadratic components, but not cubic components); the smoothed association (Lowess smoothing) between age and the Ladder is shown in Fig 1. The Ladder score increases from about 6.6 at age 40 to 7.6 at age 69, or a shift of 1 scale-point (0.52 standard deviations) over these years. The shape and level of the curve is slightly different than that reported in 2010 for the ages studied (those in their early 40s were relatively lower in the current sample), but the increase in later age is consistent with the prior literature.

### Demographic and explanatory variables: Associations with age and the Ladder

As previously mentioned, in order to reduce the age-curve a demographic or explanatory variable must be associated with both age and with the Ladder in the same direction [27]. Regarding associations with age, given the curvilinear shape of the age-Ladder association, regressions evaluating linear and quadratic components were evaluated and showed significant associations with all variables except Male, White, Future Time Perspective: Limited, and Positivity Effect (for brevity, only linear components of these associations are shown in Table 3 and these are consistent with the full regression results). College Graduate was weakly, negatively associated with age, as was income. Being married was positively associated with age. Turning to the explanatory variables, the strongest associations were for Family Strain (decreasing over age), Wisdom (increasing), Distress-Depression (decreasing), Friend Strain (decreasing), and Social Comparison (decreasing). These associations are generally consistent with Carstensen's theory. However, contrary to expectations, larger Network Size and Future Time Perspective: Open were positively associated with older age.

Regarding associations with the Ladder, College Graduate, Married, and Income all showed positive associations, whereas Male and White did not. Relatively strong associations were found for the explanatory variables and the Ladder: Perceived stress, Distress-Depression, Family Strain, Friend Strain, and Future Time Perspective: Limited were negatively associated, whereas Future Time Perspective:Open, Network Size, Social Satisfaction, Family Support, Friend Support, and Wisdom were positively associated with the Ladder. These are all highly plausible associations.

In summary, most, but not all, of the explanatory variables had associations with *both* age and SWB outcomes in the same direction, therein allowing the possibility that they could explain or affect the age-Cantril curve.

Table 2. Descriptive statistics (N = 3,294).

| Variable | Mean | SD |
|---|---|---|
| Age | 54.84 | 8.56 |
| Cantril Ladder | 6.85 | 1.89 |
| Perceived Stress | 2.36 | 0.76 |
| Distress-Depression | 1.85 | 0.94 |
| Future time perspective as open | 3.83 | 0.81 |
| Future time perspective as limited | 3.09 | 0.89 |
| Social comparison | 2.91 | 0.63 |
| Social network size | 35.48 | 38.96 |
| Social satisfaction | 3.79 | 0.88 |
| Support from family | 4.41 | 0.85 |
| Support from friends | 4.36 | 0.76 |
| Strain from family | 2.62 | 1.17 |
| Strain from friends | 2.35 | 1.07 |
| Wisdom | 3.39 | 0.54 |
| Positivity Effect | 0.13 | 0.22 |
| **Measure** | **n** | **%** |
| Sex | | |
| Female | 1708 | 51.9 |
| Male | 1586 | 48.1 |
| Race/ethnicity | | |
| Non-white | 694 | 21.1 |
| White | 2600 | 78.9 |
| Education | | |
| Some college or less | 1521 | 46.2 |
| Bachelor's degree or more | 1773 | 53.8 |
| Marital Status | | |
| Married | 2084 | 36.7 |
| Not married | 1210 | 63.3 |
| Household Income | | |
| $4,000 | 72 | 2.2 |
| $7,500 | 54 | 1.6 |
| $12,500 | 86 | 2.6 |
| $20,000 | 250 | 7.6 |
| $30,000 | 278 | 8.4 |
| $42,500 | 366 | 11.1 |
| $62,500 | 674 | 20.5 |
| $87,500 | 553 | 16.8 |
| $125,000 | 961 | 29.2 |

## Do the demographic and explanatory variables account for the curve?

Tests of the primary hypothesis that associations between age and the Ladder are partly explained by the explanatory variables were based on four regression models (see Methods) where the final model included the explanatory variables after controlling for age and demographic variables. As hypothesized, significant changes in the age regression coefficients were observed as the models included greater numbers of predictors. The curve with age is shown in Model 1 (see Fig 2) and is the starting point for testing the effect of the addition of other

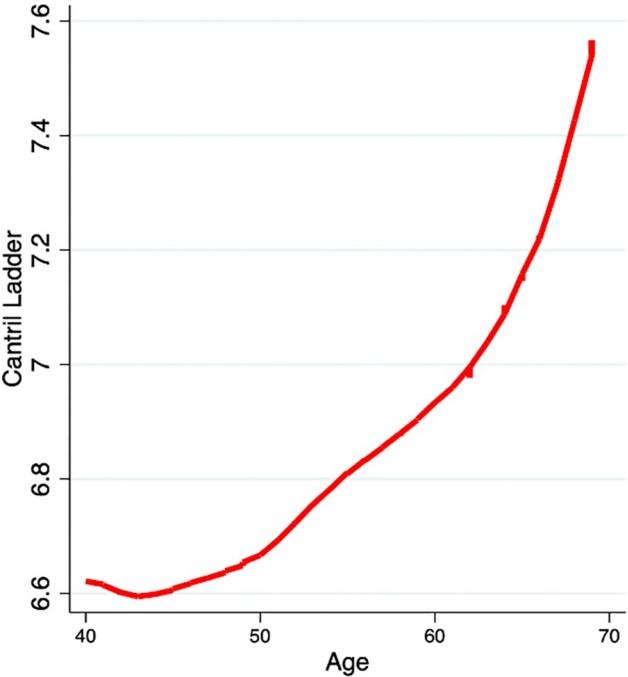

**Fig 1. Lowess smoothed curve for age and the Cantril Ladder.**

demographic and explanatory variables. Model 2 added three stable demographic variables (Sex; College Graduate; and, White) and there was a significant, if barely detectable shift in the curve (Model 1->Model 2; $X^2(2)$ = 15.4, p < .001). Model 3 added two demographic variables that were more likely to vary with age (Married and Income), yet they did not significantly influence the age-Ladder curve (Model 2->Model 3; $X^2(2)$ = 2.9, p = n.s.) The largest shift in the age curves occurred in Model 4 when the 13 explanatory variables entered the model (Model 3->Model 4; $X^2(2)$ = 80.8, p < .0001). This is seen clearly in Fig 2: there is considerable flattening of the curve compared with the other three models. Overall, at the median age for the sample (56 years), the 13 explanatory variables together flatten the linear age slope by 68.6%, after controlling for the demographic variables.

To understand the contribution of each explanatory variable, additional regression analyses were conducted. Given the moderate correlations among the explanatory variables, each variable was evaluated separately. The extent to which each explanatory variable reduces the (linear) effect of age on the Ladder is shown in Table 4. Several variables substantially reduce the age coefficient: Perceived Stress has the largest effect (84%), followed by Distress-Depression (58%), Future Time Perspective:Open (37%), Wisdom(33%), Social Satisfaction (29%), and Family Strain (26%). It was possible for a single explanatory variable to account for a greater proportion of the total age effect on the Ladder compared to the model with all 13 explanatory variables entered simultaneously, given that some variables in the latter model acted as suppressor variables in concert with other variables.

## Discussion

A previously unexplained "paradoxical" increase in SWB with older age observed in many studies was substantially accounted for by age-related patterns of social relationships, lower

**Table 3. Correlation coefficients for the Cantril Ladder, age, and explanatory variables.**

| Variable | Correlation with Age | Correlation with Cantril Ladder |
|---|---|---|
| Cantril Ladder | .123*** | |
| Male | .024 | .028 |
| Education | -.037* | .164*** |
| White | -.012 | .002 |
| Married | .055*** | .261*** |
| Log Income | -.029** | .359*** |
| Perceived Stress | -.217*** | -.572*** |
| Distress-Depression | -.171*** | -.537*** |
| FTP Open | .102*** | .548*** |
| FTP Limit | -.006 | -.327*** |
| Soc Comparison | -.154*** | -.042* |
| Networks | .042* | .180*** |
| Social Satisfaction | .102*** | .444*** |
| Family Support | .051** | .309*** |
| Friend Support | .051** | .290*** |
| Family Strain | -.217** | -.214*** |
| Friend Strain | -.168*** | -.161*** |
| Wisdom | .181*** | .318*** |
| Positivity Effect | -.023 | -.017 |

N = 3,294

*p < .05

**p < .01

***p < .001.

depression, and cognitive states. Considering that all of the explanatory variables examined in this study were measured with some degree of error, accounting for two-thirds of the observed curve is, in our opinion, substantial. To be sure, we are not arguing that the findings provide a causal explanation given the limited reach of cross-sectional data. Rather, we suggest that the explanatory variables studied here *may* be part of a developmental process that results in the age-Ladder curve. As such, we recommend that longitudinal associations be examined to understand the temporal changes amongst the measures to determine if shifts in the explanatory variables precede changes in subjective well-being, which would imply a possibly causal interpretation.

Regarding the importance of particular explanatory variables and their associations with age and the Ladder, results were partially confirmatory of Socioemotional Selectivity Theory. As predicted by the theory, higher levels of satisfaction with social relationships in older age explained some of the increase in well-being. Likewise, perceptions of strain from family and friends was lower in older age and partially explained the well-being increase. Yet there were also findings that were not consistent with the SST. First, an index of the Positivity Effect was not associated with age and, therefore, could not influence the curve. A plausible explanation for this null finding is that the range of age studied here was not wide enough for this effect to emerge. Many of the prior findings of the Positivity Effect have contrasted relatively younger groups (i.e., in their 20s) with relatively older groups, whereas this study did not include younger or very old individuals. Second, departing from the SST predictions of smaller social networks in older age, Social Network size did not increase with age and did not influence the

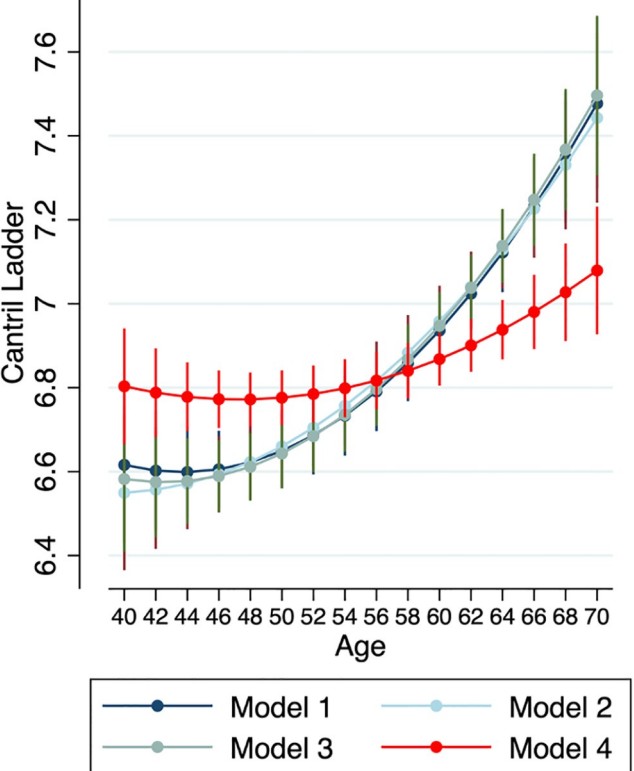

**Fig 2. Predicted curves for the Cantril Ladder for each of the four regression models.**

**Table 4. Proportion of linear age effect on the Cantril Ladder accounted for by each of the explanatory variables.**

|  | Proportion reduction of age effect | 95% confidence interval |
|---|---|---|
| Perceived Stress | .837 | .708 –.971 |
| Distress-Depression | .584 | .474 –.736 |
| FTP Open | .376 | .275 –.490 |
| FTP Limited | -.015 | -.100 –.053 |
| Social Comparison | .037 | .004 –.078 |
| Network Size | .029 | .0004 –.064 |
| Social Satisfaction | .292 | .203 –.401 |
| Fam Support | .094 | .043 –.158 |
| Friend Support | .119 | .065 –.185 |
| Fam Strain | .256 | .188 –.352 |
| Friend Strain | .144 | .092 –.215 |
| Wisdom | .332 | .256 –.442 |
| Positivity Effect | .001 | -.003 –.012 |
| N = 3,294 |  |  |

Note: 95% confidence intervals were constructed using a bias corrected bootstrap procedure with 10,000 bootstrap resamples.

shape of the age-Ladder curve. It is possible that our modification of Social Network procedures for Internet administration were not successful, and may be subject to reporting biases. Finally, a central tenet of SST is that with advancing age, individuals perceive the future as more limited and less open. Our unexpected finding that older age is associated with greater openness in perceptions of the future may reflect the use of a newly validated instrument that was developed in response to long-standing calls for a multidimensional assessment of FTP (rather than a more widely used unidimensional instrument).

Of particular interest was the substantial ability of both perceived stress and distress-depression to reduce the age-Cantril gradient when considered as single predictors. Each of these variables was correlated between .5 and .6 with the Ladder, confirming that there may be considerable conceptual overall between stress, distress-depression, and well-being. This makes intuitive sense as it is difficult to imagine someone with high levels of stress or distress-depression reporting that their lives are quite good. Nevertheless, knowing that the age-gradient is greatly diminished by controlling for these negative states can inform our interpretation of the gradient and highlights the importance of consider the interplay among variables that are often considered in isolation.

Higher levels of Wisdom in older age also partially accounted for the age-Ladder curve, although not to the same extent as perceived stress or distress-depression. While not an explicit component of Socioemotional Selectivity Theory, this result is concordant with other theories about the salutary effects of knowledge and insights gained with increasing age [28, 29]. It is also notable that reductions in distress-depression, family strain, and friend strain were associated with flattening the curve. Identification of psychological or social characteristics that may play a role in subjective well-being levels allow for the possibility to harness or target them in interventions and educational programs. Confirming these and the other associations with appropriate analyses of longitudinal data at a minimum should precede any policy or interventional efforts.

The interpretation of these results would be incomplete without discussing commonalities among the explanatory variables and the Cantril Ladder. Most of the variables are self-report measures of subjective states and, as such, likely share method variance (e.g., that contextual effects associated with test taking that may influence all of the self-reports). As mentioned above, the explanatory variables also may be viewed as having significant conceptual overlap with the Ladder, which would likely enhance their ability to explain the relationship between age and the Ladder. Nevertheless, these issues are inherent when testing hypotheses that are based upon one psychological construct influencing others in cross-sectional survey studies and are a limitation of these analyses. We also acknowledge that cohort effects may be influencing the age-Ladder gradient and should be considered in future studies.

Despite the relative flattening of the age-Ladder curves shown above, it is also the case that the age curve has not been entirely eliminated by the explanatory variables. Other factors not considered here are likely impacting SWB over the lifespan, and these deserve research attention. Biological and environmental variables are associated with aging and one may speculate that they are involved in the residual association between age and the Cantril Ladder [30]. Additionally, it is possible that other psychological processes contribute to the gradient such as age-related differential standards for judging one's satisfaction with life, although recent work has suggested this may not be the case [31]. Overall, we believe that this work contributes to a more complete understanding of the potential roles of social and psychological factors in well-being over the lifespan.

## Author Contributions

**Conceptualization:** Arthur A. Stone, Joan E. Broderick.

**Data curation:** Arthur A. Stone, Diana Wang.

**Formal analysis:** Arthur A. Stone, Stefan Schneider.

**Funding acquisition:** Arthur A. Stone, Joan E. Broderick.

**Investigation:** Arthur A. Stone, Joan E. Broderick, Diana Wang, Stefan Schneider.

**Methodology:** Arthur A. Stone, Joan E. Broderick, Diana Wang, Stefan Schneider.

**Project administration:** Diana Wang.

**Resources:** Arthur A. Stone.

**Software:** Stefan Schneider.

**Supervision:** Arthur A. Stone, Joan E. Broderick.

**Visualization:** Arthur A. Stone, Stefan Schneider.

**Writing – original draft:** Arthur A. Stone.

**Writing – review & editing:** Arthur A. Stone, Joan E. Broderick, Diana Wang, Stefan Schneider.

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
