## [Decision Letter · Decision Letter 0]

30 Sep 2020

PONE-D-20-06409

Age Patterns in Subjective Well-Being are Partially Accounted for by Psychological and Social Factors Associated with Aging

PLOS ONE

Dear Dr. Stone,

Thank you for submitting your manuscript to PLOS ONE. After careful consideration, we feel that it has merit but does not fully meet PLOS ONE’s publication criteria as it currently stands. Therefore, we invite you to submit a revised version of the manuscript that addresses the points raised during the review process.

We look forward to receiving your revised manuscript.

Kind regards,

Geilson Lima Santana, M.D., Ph.D.

Academic Editor

PLOS ONE

Journal Requirements:

'AAS is a Senior Scientist with the Gallup Organization and is a consultant for Adelphi Values.'

a. Please confirm that this does not alter your adherence to all PLOS ONE policies on sharing data and materials, by including the following statement: "This does not alter our adherence to  PLOS ONE policies on sharing data and materials.” (as detailed online in our guide for authors http://journals.plos.org/plosone/s/competing-interests).  If there are restrictions on sharing of data and/or materials, please state these.

Please note that we cannot proceed with consideration of your article until this information has been declared.

Reviewers' comments:

Reviewer's Responses to Questions

**Comments to the Author**

1. Is the manuscript technically sound, and do the data support the conclusions?

Reviewer #1: Yes

2. Has the statistical analysis been performed appropriately and rigorously? 

Reviewer #1: Yes

3. Have the authors made all data underlying the findings in their manuscript fully available?

Reviewer #1: No

4. Is the manuscript presented in an intelligible fashion and written in standard English?

Reviewer #1: Yes

5. Review Comments to the Author

Reviewer #1: This is a well-written manuscript examining a number of psychological and social factors and their potential to account for the U-shaped association between age and subjective well-being. On the cover page, the authors indicate funding in the Financial Disclosures section, but the requested information is not provided in the following section. Several other sections of the cover page are similarly left blank (perhaps this is intentional, but it appears incomplete). The data availability statement seems vague and incomplete, per my reading of the instructions. The statistical methods are appropriate and study limitations are appropriately addressed.

Following is a list of questions/critiques for the authors:

• There is a type in Table 1 – the “Relationship quality: Strain from friends” is described as “Same as above but for family.”

• Initially I wanted to know why 40 was the lower age for study eligibility. This was explained later in the manuscript, and as the authors state, may serve to dampen the findings

• Was the median income individual income or household income?

• In the methods section, the authors state that “sociodemographic criteria were set to fill predetermined quota of …race/ethnicity that corresponded to Census breakdown” (i.e., 64% white/Caucasian). However, in the results section it is stated that the sample is 79% white. How do the authors account for this discrepancy and how might it be affecting the results?

• The authors point to the unexpected finding that larger network size and future time perspective: open were positively associated with older age. Why might this be the case?

6. PLOS authors have the option to publish the peer review history of their article (what does this mean?). If published, this will include your full peer review and any attached files.

Reviewer #1: No

---

## [Author Response · Author response to Decision Letter 0]

20 Oct 2020

Dear Dr. Santana:

Thank you for your consideration of our manuscript and thoughtful feedback. We have incorporated the reviewer’s suggestions throughout the manuscript, and provide comments in response to each point below. 

- We have uploaded the dataset and documentation onto Open Science Framework and included the link to access in the submission portal. We have addressed the blank portions of the cover page and thank the reviewer for bringing this to our attention. The financial disclosures have been added to the submission portal. 

- Wang, D. (2020, October 15). Roybal Study on Subjective Well-being. Retrieved from osf.io/ufmgh

'AAS is a Senior Scientist with the Gallup Organization and is a consultant for Adelphi Values.'

a. Please confirm that this does not alter your adherence to all PLOS ONE policies on sharing data and materials, by including the following statement: "This does not alter our adherence to PLOS ONE policies on sharing data and materials.” (as detailed online in our guide for authors http://journals.plos.org/plosone/s/competing-interests). If there are restrictions on sharing of data and/or materials, please state these.

Please note that we cannot proceed with consideration of your article until this information has been declared.

- AAS is a Senior Scientist with the Gallup Organization and is a consultant for Adelphi Values. This does not alter our adherence to PLOS ONE policies on sharing data and materials.

Reviewer #1: This is a well-written manuscript examining a number of psychological and social factors and their potential to account for the U-shaped association between age and subjective well-being. On the cover page, the authors indicate funding in the Financial Disclosures section, but the requested information is not provided in the following section. Several other sections of the cover page are similarly left blank (perhaps this is intentional, but it appears incomplete). The data availability statement seems vague and incomplete, per my reading of the instructions. The statistical methods are appropriate and study limitations are appropriately addressed.

Following is a list of questions/critiques for the authors:

There is a typo in Table 1 – the “Relationship quality: Strain from friends” is described as “Same as above but for family.”

- Thank you for pointing this out. We have corrected this typo in Table 1.

Initially I wanted to know why 40 was the lower age for study eligibility. This was explained later in the manuscript, and as the authors state, may serve to dampen the findings

- We agree that the reason could be clarified in the introduction, and have included an explanation on lines 32-34.

Was the median income individual income or household income?

- The measure for income was household income: “Which of the income groups represents your total combined household income, before taxes, for the past 12 months? Include income from all sources such as wages, salaries, social security or retirement benefits, help from relatives, rent from property and so forth.”

- We have made clarified this in the paper. Please see lines 109 and 157.

In the methods section, the authors state that “sociodemographic criteria were set to fill predetermined quota of …race/ethnicity that corresponded to Census breakdown” (i.e., 64% white/Caucasian). However, in the results section it is stated that the sample is 79% white. How do the authors account for this discrepancy and how might it be affecting the results?

- Thank you for bringing this to our attention. Our reporting of the targeted race/ethnicity breakdown of the demographic criteria was not clear, and we have corrected this. 

- To clarify, we set recruitment criteria to 64% White/Non-Hispanic and 16% Hispanic/Latino according to Census breakdown. 

- In the resulting sample achieved in the study, 66.5% of participants are White and not Hispanic/Latino; a total of 78.8% were White and Hispanic. Our intention in the analysis was to distinguish between White (inclusive of those who are also Hispanic/Latino) and non-White as is standard in much of the subjective well-being literature, so we used a dichotomous indicator of White and Non-White.

- Please see edits in lines 88 and 140.

The authors point to the unexpected finding that larger network size and future time perspective: open were positively associated with older age. Why might this be the case?

- These unexpected findings may be due to a number of factors including the choice of measures and the age range of participants used in the study. Previous studies investigating future time perspective used a unidimensional scale that assessed a continuum from open future on one end to limited future on the other end. However, factor analytic research shows that there are actually two separate factors, and an increase in one is not necessarily associated with a decrease in the other. Thus, it is possible that with a multi-dimensional scale used in this study that open FTP does not decrease with older age. Furthermore, other studies using this measure show that the largest differences are between younger adults and older adults. Our use of a middle-age and older adult sample could be another reason for this unexpected finding. Please see changes in the discussion on lines 265 – 269.

- Larger network size with advancing age is unexpected because previous work has demonstrated shrinking networks with advancing age. This may be due to our adaptation of the social network size measure into a web-based format for administration for an online study. In previous work, Antonucci’s concentric circles measure of social network size required respondents to fill out names of individuals who they consider to be in the innermost, middle, and outer circles. In addition, respondents are typically asked to report structural and functional characteristics of the first 10 people listed in the network – age, sex, relationship to the respondent, time elapsed since start of relationship, frequency of contact, etc.). Some of these features could not be translated onto a web-based format and we recognize that the adapted version may have been subject to reporting biases. This is a limitation that we address in line 260-264.

We hope you find these modifications to the paper acceptable and we look forward to hearing from you soon.

---

## [Editor Report · Decision Letter 1]

9 Nov 2020

Age Patterns in Subjective Well-Being are Partially Accounted for by Psychological and Social Factors Associated with Aging

PONE-D-20-06409R1

Dear Dr. Stone,

We’re pleased to inform you that your manuscript has been judged scientifically suitable for publication and will be formally accepted for publication once it meets all outstanding technical requirements.

Kind regards,

Geilson Lima Santana, M.D., Ph.D.

Academic Editor

PLOS ONE
---

## [Editor Report · Acceptance letter]

19 Nov 2020

PONE-D-20-06409R1 

Age Patterns in Subjective Well-Being are Partially Accounted for by Psychological and Social Factors Associated with Aging 

Dear Dr. Stone:

I'm pleased to inform you that your manuscript has been deemed suitable for publication in PLOS ONE. Congratulations! Your manuscript is now with our production department. 

Kind regards, 

on behalf of

Dr. Geilson Lima Santana 

Academic Editor

PLOS ONE